# Cyclic Performance of End-Plate Biaxial Moment Connection with HSS Columns

**Eduardo Nuñez [1],\*** , **Roberto Lichtemberg [2]** and **Ricardo Herrera [2]**

[1]  Department of Civil Engineering, Universidad Católica de la Santísima Concepción, Concepcion 3349001, Chile
[2]  Department of Civil Engineering, Universidad de Chile, Santiago de Chile 8370449, Chile;
    roberto.lichtemberg@ug.uchile.cl (R.L.); riherrer@ing.uchile.cl (R.H.)
\*  Correspondence: enunez@ucsc.cl; Tel.: +56-9-51277382

**Abstract:** This paper presents a numerical study on the seismic performance of end-plate moment connection between I-beam to HSS (hollow structural section) column stiffened by outer diaphragms (EP-HSS). In previous experimental research, this moment connection showed a satisfactory performance according to requirements established in Seismic provisions. However, one type of joint was studied and bidirectional and axial loads were not considered. In this since, several configurations representative of 2D interior joints and 3D interior and exterior joints in a steel building were modeled and subjected to unidirectional or bidirectional cyclic displacements according to protocol in seismic provisions. Firstly, a similar joint configuration was calibrated from experimental data, obtaining an acceptable adjustment. The assessment of seismic performance was based on hysteretic curves, failure mechanisms, stiffness, dissipated energy, and equivalent damping. The results obtained showed a ductile failure modes for 2D and 3D joint configurations with EP-HSS moment connection. The axial load has no significant effect on the moment connection. However, it affects the column strength due to the increase of the stresses in the column wall. Compared with 2D joints, 3D joints reached higher deformations even when a similar number of beams is used. The external diaphragms to the column panel zone provided rigidity in the joints and no degradation of slope for each loop in load/reload segment for elastic loop; therefore, curves without pinching were observed. All inelastic deformation is concentrated mainly in the beams. A moment resistance above 80% of the capacity of the beam at a drift of 4% is achieved in all joints. From the results reached, the use of EP-HSS moment connection with hollow structural section columns is a reliable alternative in seismic zones when steel moment frames are employed.

**Keywords:** biaxial resistance; seismic performance; bolted moment connections; finite element method; steel structure; hollow structural section; seismic design

---

## 1. Introduction

Steel structures have shown an acceptable performance under seismic loads, especially in those systems with structural redundancy, compact members, and connections designed for the expected capacity of their members. Steel moment resisting framed buildings are advantageous when long spans are required (i.e., parking and office buildings). In US practice, rolled wide flange sections are commonly used as columns, using moment frames connected to strong axis of the columns and braced frames in weak axis of the column (because it is difficult to provide a strong column–weak beam scheme when the girder frames into the weak axis of the column). Tubular sections have similar moments of inertia for both principal axes, making them better suited to resist biaxial moments than wide flange sections. Additionally, tubular sections have a higher lateral torsional stiffness than wide flange sections, requiring fewer, if any, lateral bracing points, according to [1,2]. Recently, during the

large 2011 Tohoku Japan earthquake, special moment frames with HSS (hollow structural section) columns showed a favorable seismic performance, with respect to other structural configurations [3,4]. In moment resisting frames, the seismic performance of beam-to-column connections has a direct incidence in the resistance, stability and stiffness of the structure.

Numerous studies have focused on welded and bolted connections between wide flange beams and tubular columns, tubular beams and tubular columns, and wide flange beams and wide flange columns. A brief summary of the works more relevant to this research follows.

A numerical and experimental study on HSS members was conducted by [5–7]. The cyclic response of HSS-to-HSS moment connections was studied, proposing a welded connection incorporating plates that allowed an enhanced performance ensuring the dissipation in the beam. However, the connection was fully welded, requiring the use of field welding. A numerical study of ConXtech® ConXL™ moment connection in box columns not filled was performed by [8]. Results showed a suitable seismic behavior with axial force in single and biaxial loading until 0.04 radian rotations.

A moment connection with reduced beam section (Tubular Web-Reduced Beam Section connection, TW-RBS) was studied by [9]. The results showed a reduction in the flexural capability of the beam and a ductile behavior of the beam-to-column connection. A bidirectional effect was not considered. A numerical study on the ConXtech® ConXL™ moment connection was conducted by [10], using the finite element method, considering bidirectional effects. The study included concrete filled tubular and HSS columns. Numerous joint configurations without concrete infill in the column showed inelastic behavior in beam for axial loads up to 40% of the capacity of the column. For higher values, a failure mechanism appeared in the column. Additionally, Ref. [11] proposed a bolted moment connection between an I-beam and an H-column for high-rise steel structures. Beam and column are connected through a bolted internal diaphragm the hysteretic behavior showed an influence of slip in cover plates. However, the connection can sustain a rotation of 0.04 radians.

A numerical and experimental research on a new moment connection with HSS column was studied by [12]. This moment connection joins I beams with an HSS column through a bolted end-plate and outer diaphragms. The thickness of the end-plate is reduced 16% using a new yield line pattern, while maintaining a seismic performance as required according to [13]. This reduction is a result obtained from numerical study performed and validated by experimental tests. The results showed that End Plate to Hollow Structural Section (EP-HSS) connection reached 5% drift, a ductile failure mechanism in the beam rather than the column and hysteretic behavior without brittle failure. The study did not include bidirectional effects or axial loading in the column.

The research performed by [14], studied the cyclic response of built-up box columns connected to I beams using the four-bolt extended end-plate connection, subjected to bidirectional bending and axial load on the column. The results revealed that the failure is concentrated in the beams of all joint configurations except for the columns with axial load equal to 75% of the column capacity, where a combined failure mechanism is achieved. The energy dissipation capacity of joints with a greater number of beams is lower than joints with fewer beams. However, the thickness end-plate was designed according to [13]; therefore, it was not optimized and hollow structural sections (HSS) was not studied.

A numerical and experimental research conducted by [15] studied a moment connection to square HSS column with blind bolts. The results showed that beam size and end-plate thickness can improve the stiffness and moment capacity of the connection. Cyclic response was not evaluated. Furthermore, Ref. [16] studied a typical moment connection in cold-formed steel. The results showed that using bolting friction-slip mechanism improve the energy dissipation capacity of the connection. A new moment connection with double-through plates to concrete filled steel tube was studied by [17]. The results showed that the connection exhibited similar performance to single plate connections (pinned behavior).

A similar moment connection according to the research performed by [11] was conducted by [18]. In this opportunity, a seismic performance of bolted connection between I-beam and H-column with

web end-plate was studied as extension of previous research. The results obtained revealed that the proposed connection had an acceptable hysteretic behavior. According to [13], 11 connections are prequalified for seismic design of steel moment frames; from these connections only one type (ConXtech® ConXL$^{TM}$ moment connection, proprietary and protected by patent) could be used with HSS columns if it is concrete filled, with RBS beams connected to the column through collar elements.

In summary, researches has focused on: (a) welded and bolted moment connections with wide flange beams to wide flange columns, (b) welded moment connections with wide flange beams to tubular columns, and (c) welded connections with tubular beams to tubular columns. However, the bidirectional effect, number of connected beams and variability were not considered in moment connections and only the ConXtech® ConXL$^{TM}$ moment connection (patented) can be used.

In this research, a numerical study of end-plate to hollow structural section (EP-HSS) moment connection is performed to evaluate the cyclic behavior of 2D and 3D joints using the Finite Element Method (FEM). The study is an extension of the research previously carried out by [12], which a new moment connection for I-beam to HSS-column was proposed from experimental and numerical study in 2D joint configuration according to Seismic provisions [19]. Unlike the study conducted by [12], 2D (with two beams) and 3D joints configurations subjected to different levels of axial loads were performed using FE models, calibrated from previous research in [12]. The response in terms of resistance and deformation is analyzed from moment rotation curves, stiffness, damping, and dissipated energy. This connection is a proposal for the use of I-beam and HSS column and 3D joints with optimized end-plate avoiding especially joints welded on site, reduction in flanges of beam and complex fabrication process for use of internal diaphragms. For this purpose, various configurations of 2D and 3D joints were studied with different load levels respect to the column yield load Py (0%, 25%, and 50%). The 2D joints are 1E (one beam in exterior plane) and 2I (two beams in interior plane). The 3D joints are 2E (two beams in exterior plane), 3E (three beams in exterior plane), and 4I (four beams in interior plane), as shown in Figure 1.

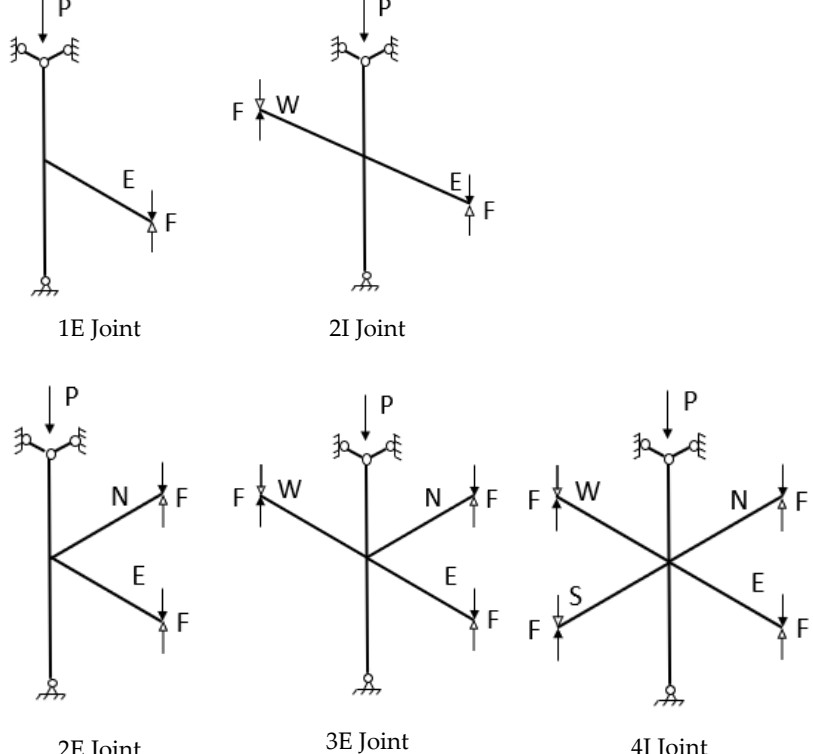

**Figure 1.** Loading conditions for 2D and 3D joints.

The EP-HSS moment connection consists of two end-plates connected by means of high strength bolts to the HSS column through external diaphragms (see Figure 2), which improves the erection process on site avoiding field welding. These advantages in addition to the biaxial resistance of HSS columns provide a reliable alternative in seismic applications. Beam and column are high ductility sections according to [19]. The size of beam and column in the joints is shown in Figure 3. A seismic design of residential building with four story levels located in Santiago, Chile, and steel moment frames in orthogonal directions was performed. Tubular columns and I-beams were specified with biaxial moment connections and full details of the design process can be found in [20]. The elements of the connection such as end-plate, bolts, horizontal diaphragms and welding are designed for the maximum probable moment of beam. The vertical diaphragm was designed for the maximum shear transferred by the beam to the column (see Figure 4). A strong-column/weak-beam criteria was calculated for all joints according to [19] using the E3-1 equation, $\sum M_{pc}^* / \sum M_{pb}^* > 1.0$, where $\sum M_{pc}^*$ is the sum of the projections of the nominal flexural strengths of the columns above and below the joint to the beam centerline with a reduction for the axial force in the column and $\sum M_{pb}^*$ is the sum of the projections of the expected flexural strengths of the beams at the plastic hinge locations to the column centerline.

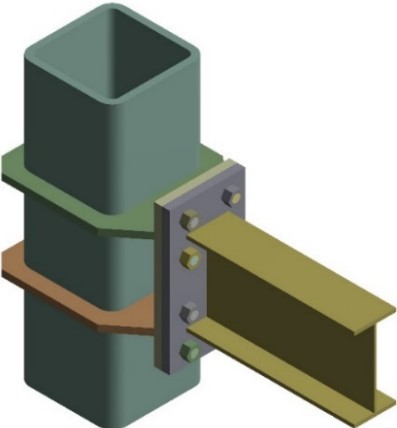

**Figure 2.** View of End Plate to Hollow Structural Section (EP-HSS) moment connection.

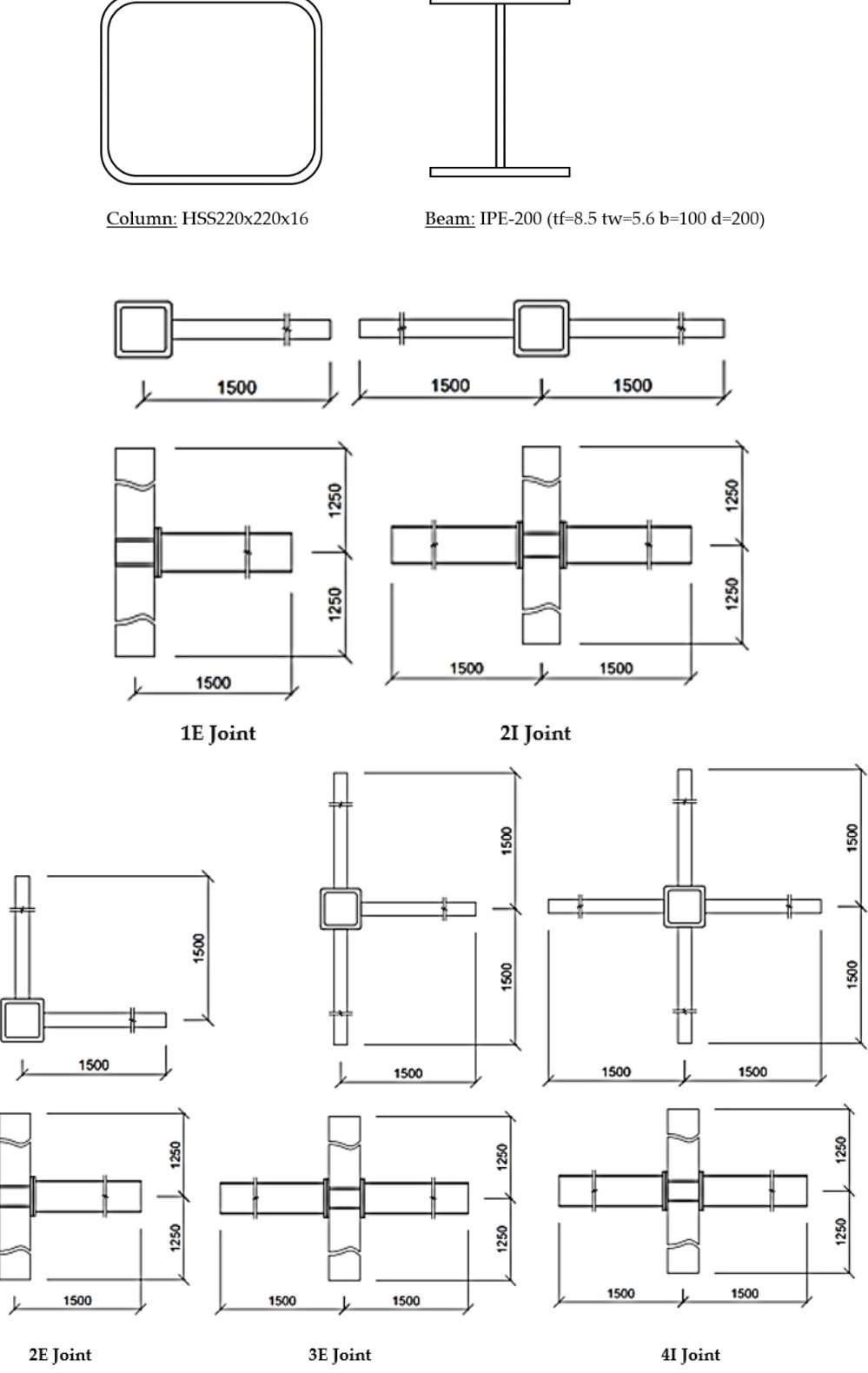

**Figure 3.** Schematic view of 2D and 3D joints [mm].

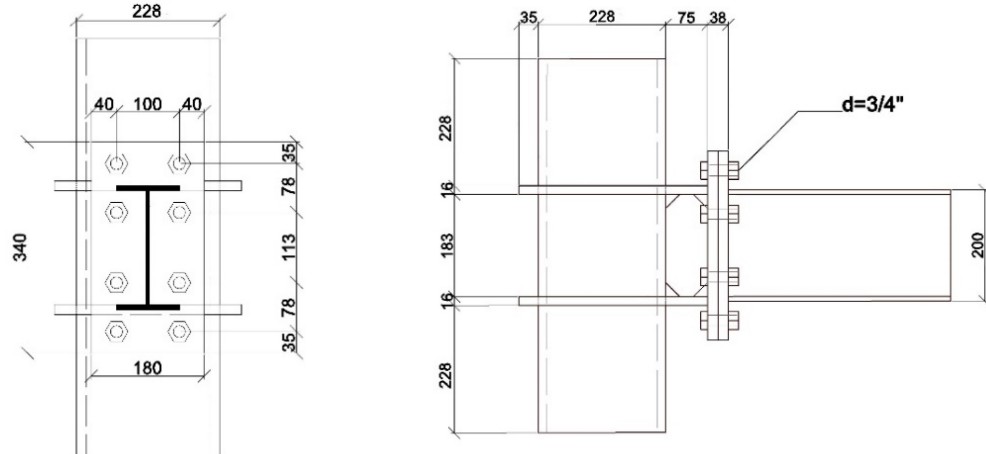

**Figure 4.** Details of end-plate [mm] and lateral view of EP-HSS moment connection [mm].

In case of joints with beams out-of-plane, an additional reduction in the flexural strengths of the column was deemed, proportional to relationship, $\sum M^*_{pb}/2Z_y$, where $M^*_{pb} =$ is the plastic flexural strength of a beam in the out-of-plane frame at the joint under consideration and $Z_y$ is the plastic section modulus of the column out of plane of the frame under consideration. In the Table 1, the ratio of the sum of moment capacity of columns to beams of each joint was reported.

**Table 1.** Strong-column/Weak-beam Moment Ratio in joints studied.

| Joint | SC/WB |
|-------|-------|
| 1E    | 4.53  |
| 2I    | 2.26  |
| 2E    | 3.59  |
| 3E    | 1.79  |
| 4I    | 1.32  |

The equivalent load-displacement method for 2D and 3D joints was used to compare the seismic performance between different configurations of steel joints according to research performed by [10,14]. In general, an equivalent force in the top column ($V_c$) is calculated by static equilibrium as the resultant force in the horizontal direction; posteriorly, equating the work performed by beam forces with work performed by the equivalent force, an equivalent displacement ($\Delta$) is calculated. The full details of the method applied in similar joints studied can be found in [14].

## 2. Finite Element (FE) Modeling

The numerical study was performed using the software ANSYS software [21] considering the constitutive laws of material, geometrics nonlinearities, contact nonlinearities, and boundary conditions. Large displacements were considered in the simulations due to the high rotation levels reached in the connections. The Incremental Newton–Raphson method was used, which the nonlinearities are considered through the sub-steps for each load step. The force convergence criterion was applied, where the residual out-of-balance force vector and the force convergence value must be below the value for convergence, according to [21]. Finally, the Augmented Lagrange method was used to reach numerical convergence in the contact zone, according to research performed by [22].

In the numerical models, general assumptions were considered as follows: the length of the column is taken as the distance between zero moment points for each case (zero moment points in columns are assumed at mid height). The welds are not included in the model considering that inelastic incursion is not expected in these elements. The diameter of the holes is assumed equal to the diameter of the bolts. This assumption is possible due to pretension applied to bolts and their performance in

the connection (bolts subjected to tension instead of bolts subjected to shear). These assumptions were verified and employed by [12]. However, if these effects need to be considered, previous studies by [23] provide a methodology to include them in the numerical model. An optimized thickness end-plate from analytical proposal by mean of yield line theory was obtained according to previous research in [12].

### 2.1. Boundary Conditions, Element Type and Loading

Similar boundary conditions to tests in experimental study conducted by [12] were applied. Consequently, the ends of columns were considered pinned supports (displacements restrained with rotations released) and vertical displacements applied to end of the beam according to loading protocol established in [19] with out-of-plane displacements restrained (see Table 2). In joint configurations with two or more beams, the loading protocol was applied simultaneously in all beams with the boundary conditions mentioned previously. As shown in the Figure 5, pretension in bolts of 70% of the nominal tension strength was applied. A "Bonded" contact was employed to simulate welding conditions. This type contact is a complete restraint of the displacements and rotations between the parts connected.

**Table 2.** Load protocol in Finite Element (FE) models, adapted from [14].

| No. | No. of Cycles | Drift Angle (θ) [rad] |
|---|---|---|
| 1 | 6 | 0.00375 |
| 2 | 6 | 0.005 |
| 3 | 6 | 0.0075 |
| 4 | 4 | 0.01 |
| 5 | 2 | 0.015 |
| 6 | 2 | 0.02 |
| 7 | 2 | 0.03 |
| 8 | 2 | 0.04 |
| 9 | 2 | 0.05 |

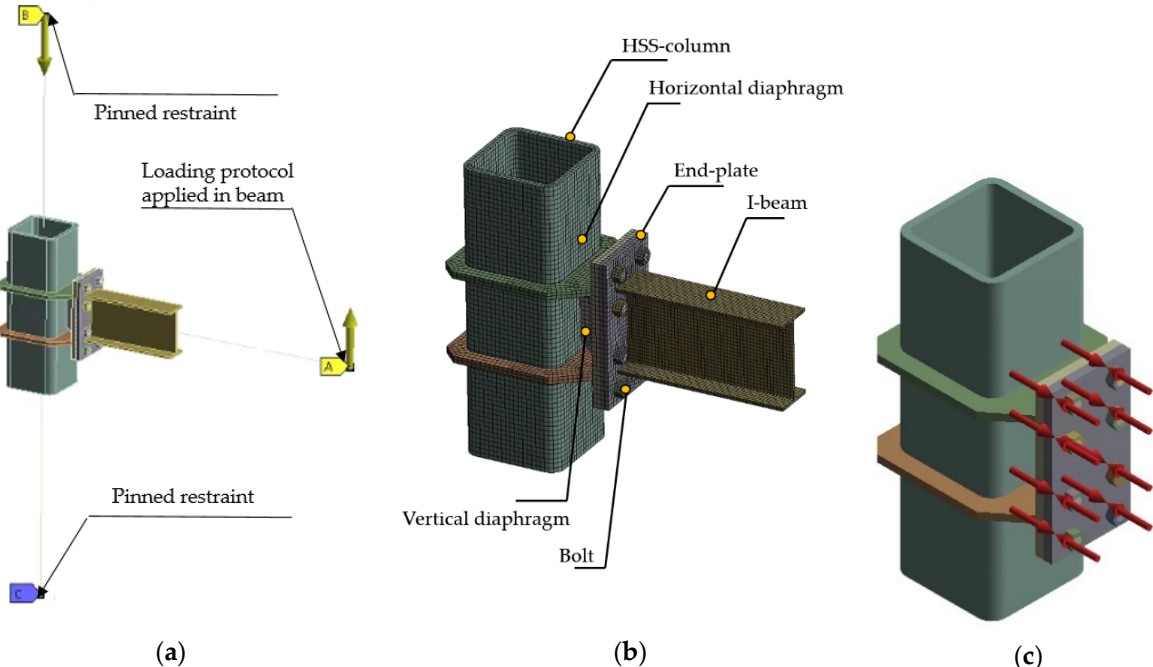

(**a**)　　　　　　　　　　(**b**)　　　　　　　　　　(**c**)

**Figure 5.** (**a**) Boundary conditions, (**b**) elements in EP-HSS connection and (**c**) bolt pretension in Finite Element (FE) model.

The interaction between elements of connection was deemed through difrent types of contacts. A "Frictional" contact with a 0.3 friction coefficient was used to simulate the contact between end plates according to [12]. Other values can be used ($\mu$ = 0.1 to $\mu$ = 1); however, variations of less than 2% in moment capacity were obtained [12,14]. Furthermore, the contact between bolts-nuts, bolts-end plate, and nuts-end plates are modelled with "Frictionless" type contacts, which allow separation between the connected parts and the tangential movement without considering the friction, following [24]. The members and plates are discretized using hexahedral and tetrahedral 3D solid elements (SOLID 185) with eight nodes with three translational degrees of freedom per node, reducing the computational effort. To reduce the computational cost and to improve the convergence, BEAM188 elements with two nodes with six degrees of freedom per node were employed, reducing the number of equations to solve due to the simplification of its formulation to represent the portion of beam and column with elastic behavior, as shown in Figure 5a. To join the BEAM188 elements to SOLID185 elements a "Bonded" contacts were used, which are a complete restraint of the displacements and rotations between the elements connected. In the Figure 6, the schematic view of 2D and 3D joints configurations studied are shown.

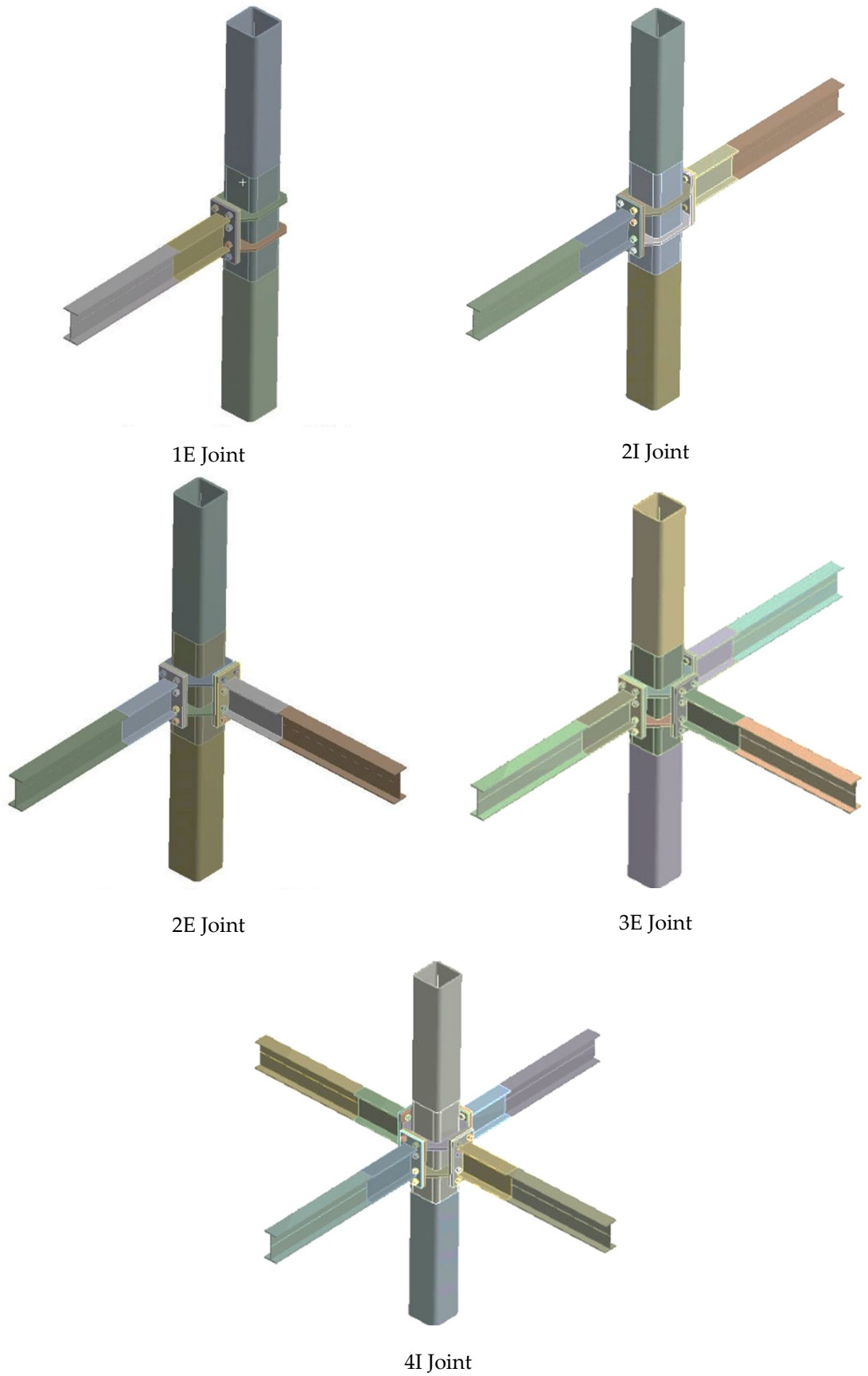

**Figure 6.** 2D and 3D Joint configurations studied.

## 2.2. Material Property

The multilinear kinematic constitutive law with the Von Mises yielding criterion was used. In this model, the yield surface remains constant in magnitude and location. However, if the specimen is first loaded and deformed in uniform tension, the load is then removed and the specimen is loaded in compression, the compressive yield stress will be less than the initial yield stress. This model is recommended for metals subjected to cyclic load according to [19]. The ASTM A36 material is assumed for beam, end plates, and horizontal and vertical diaphragms, while ASTM-A500 Gr.B and ASTM-A325 are assumed for the column and bolts, respectively. The materials were obtained from coupon tests (see Table 3) according to [12], and converted to true stress and true strain values before using them as input for the FEM models, as is shown in the Figure 7. To convert the material strain-stress curve obtained from coupon test to true stress and true strain, the following equations were used:

$$\varepsilon_{real} = \ln(1 + \varepsilon) \tag{1}$$

$$\sigma_{real} = \sigma(1 + \varepsilon) \tag{2}$$

where,

$\varepsilon$ = normal strain obtained from uniaxial tensile test.

$\sigma$ = normal stress obtained from uniaxial tensile test.

$\varepsilon_{real}$ = real normal strain.

$\sigma_{real}$ = real normal stress.

**Table 3.** Mechanical properties of test materials.

| Element | Yield Stress [MPa] | Yield Strain | Ultimate Stress [MPa] | Ultimate Strain |
|---|---|---|---|---|
| Beam, Stiffeners, End-plates | 380 | 0.0018 | 575 | 0.20 |
| Column | 496 | 0.0025 | 597 | 0.01 |
| Bolt | 634 | 0.0036 | 848 | 0.14 |

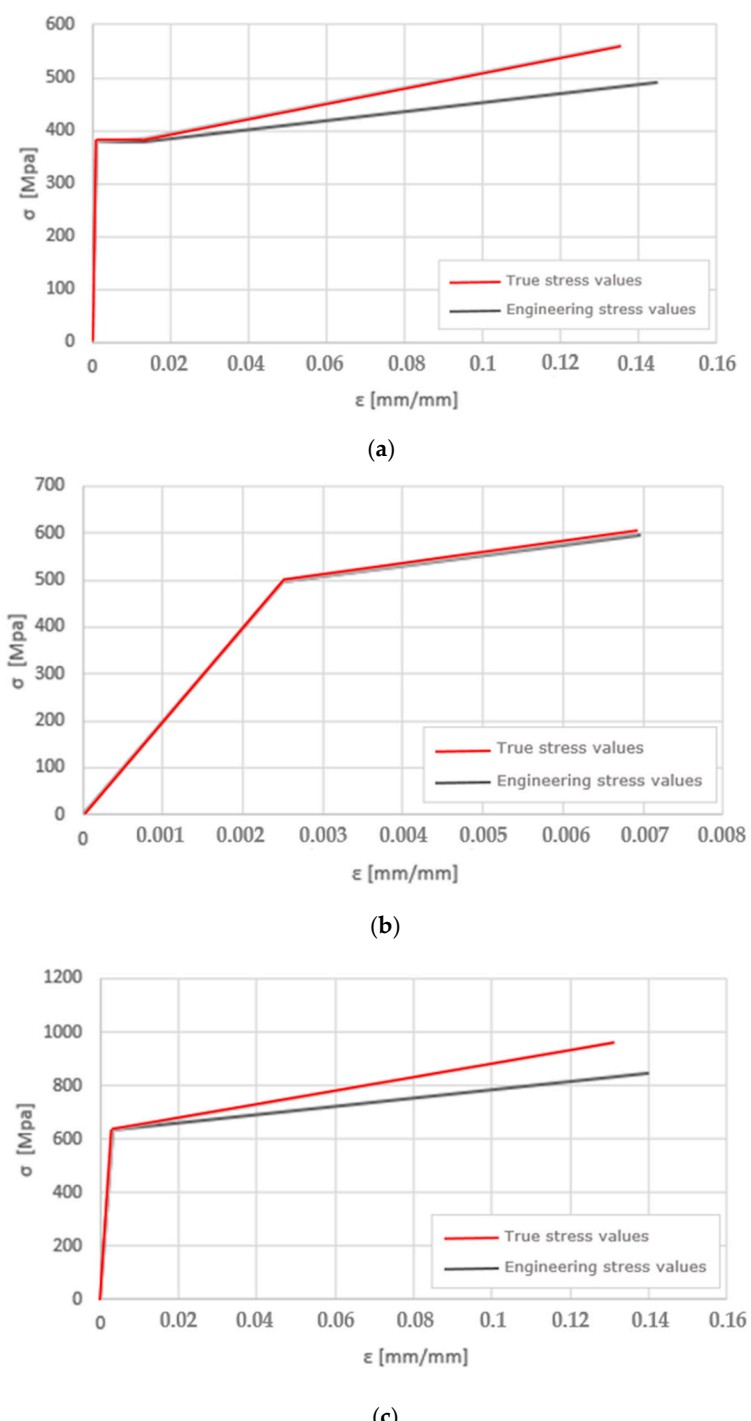

**Figure 7.** Relation stress-strain of materials used: (**a**) ASTM A36 material, (**b**) ASTM A500 Gr. B material and (**c**) ASTM A325 material.

The numerical model of 1E joint configuration was calibrated with experimental data. In the Figure 8, the normalized moment-rotation curves of tests and numerical model for 1E joint configuration were compared, achieving an acceptable adjustment. The dissipated energy comparison will be shown in Section 3.

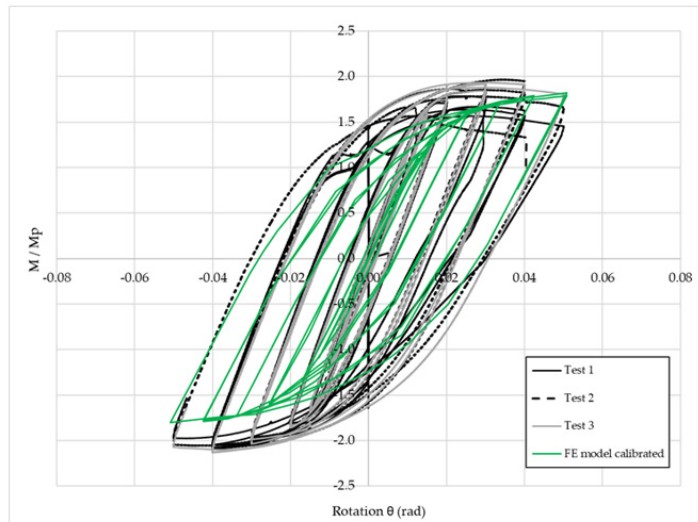

**Figure 8.** Model in FE calibrated according to [12]. Note: *Mp = Fy.Zx*, where *Mp* is the plastic moment of beam connected.

## 3. Results

The numerical study of the EP-HSS moment connection considering bidirectional effect and axial load was performed. The seismic performance of moment connections can be studied from hysteresis curves, failure modes, dissipated energy, equivalent damping ($\xi_{eq} = Ed/4\pi Eso$, where $E_d$ is the dissipated energy and $E_{so}$ is the strain energy) and stiffness, as defined in [25]. A ductile failure mechanism with plastic deformations in beams is required according to AISC seismic provisions [19]. A flexural resistance of 0.8 Mp and 4% rotation is mandatory for moment connections in seismic zones. Likewise, a combined failure mechanism (beam and column) is not desirable [19].

In this research, the performance indicators such as moment-rotation curves, stresses and plastic deformation distribution in 2D and 3D joint configurations are reported. Due to the volume of data obtained, only the east beam results are reported for all joint configurations.

Normalized moment-rotation curves are shown in Figure 9. A drift angle and flexural strength of the beam greater than 0.04 [rad] and 0.80 Mp (Mp = 43.87 [kN.m]), respectively, are obtained for joints 1E, 2E, 2I, 3E and 4I. No degradation of stiffness and resistance were observed. However, a slight pinching was reached in models with 25% and 50% of the column yield axial load with respect to 0% axial load for 2I, 3E and 4E joints. Hysteretic behavior with pinching was reported in different moment connections such as [11,17,18] with loss of resistance and stiffness by combined beam-column failure mechanisms and damage in elements of connection. For axial load cases higher than 0%, plastic strains in the column walls are reached, explaining the drop of resistance in the hysteretic behavior. However, the requirements according to [19] for all joint configurations are satisfied.

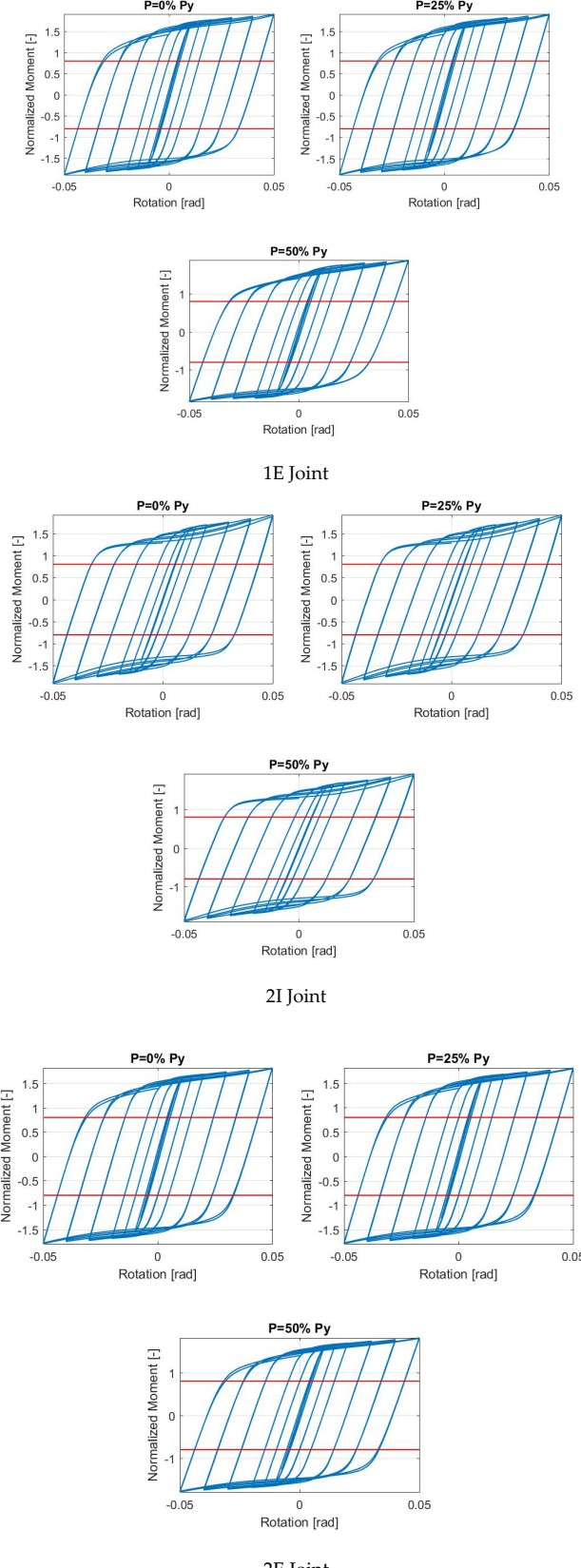

**Figure 9.** *Cont.*

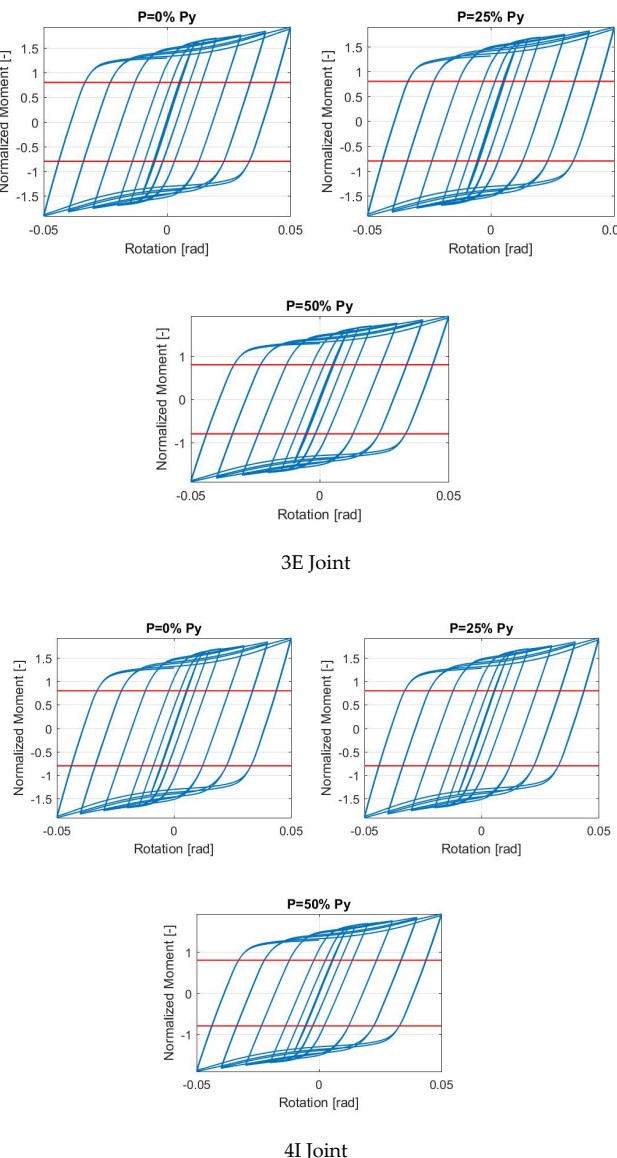

**Figure 9.** Summary of Normalized Moment-Rotation curves. Note: P, is the axial load applied in numerical model, Py = FyAg, where Fy is the plastic modulus and Ag is the gross area.

As shown in Figure 10, von Mises equivalent stresses in the beam greater than the yield stress are reached. Additionally, a higher stress concentration in the column for the models with 25% and 50% column yield axial load was obtained in comparison to the model with no axial load for all joints. This shows the influence of the level of column axial load in the different models. Plastic strains appear mainly in beams and in some diaphragms (see Figure 11). No plastic strains in the column are obtained. A ductile failure mechanism is obtained, where beams reach plastic deformation without inelastic behavior in column.

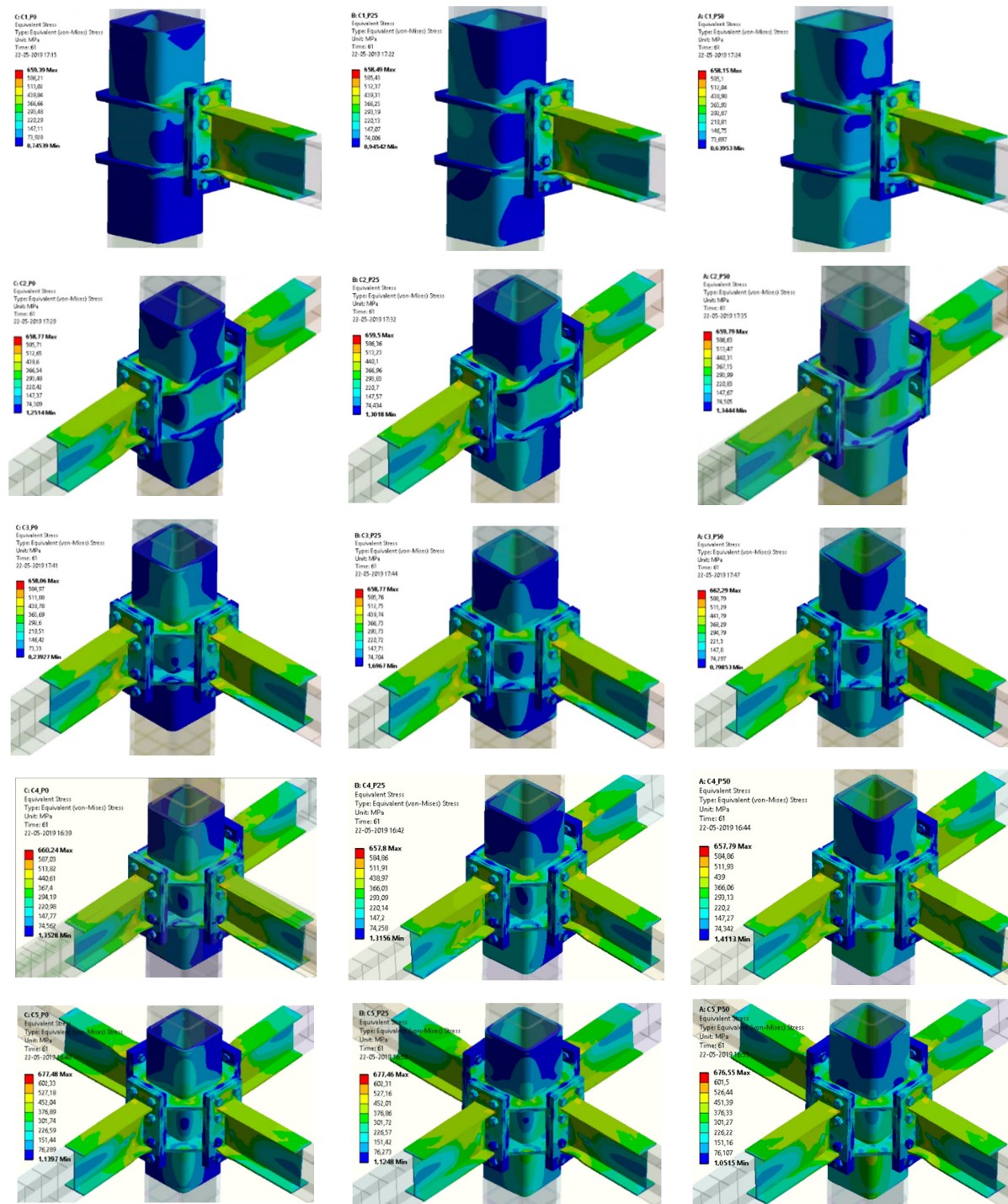

**Figure 10.** Distribution of Von Mises stresses in joints.

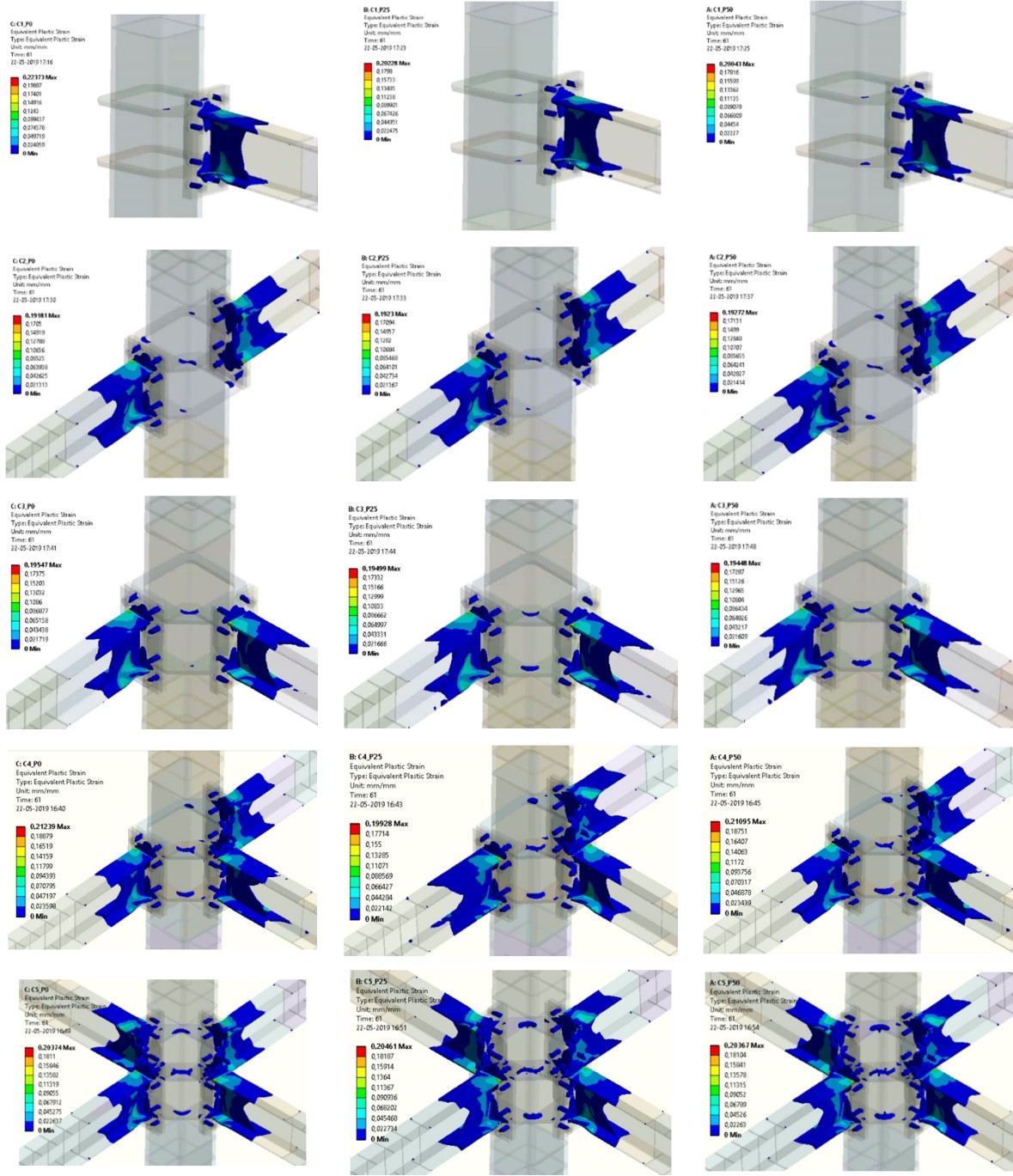

**Figure 11.** Distribution of plastic strains in joints.

A comparison of moment-rotation curve, load-displacement curve, tangent stiffness, secant stiffness and equivalent damping in 2D and 3D joints was performed employing the equivalent load method [10,14]. As shown in Figure 12, similar hysteretic behavior for all joints is obtained. The 2I and 2E joints have the same number of elements; however, lower equivalent resistance and higher equivalent displacement in the joint 2E is obtained. Consequently, displacement and rotation levels in 3D joints are greater than 2D joints. In general, the joints can reach 5% drift.

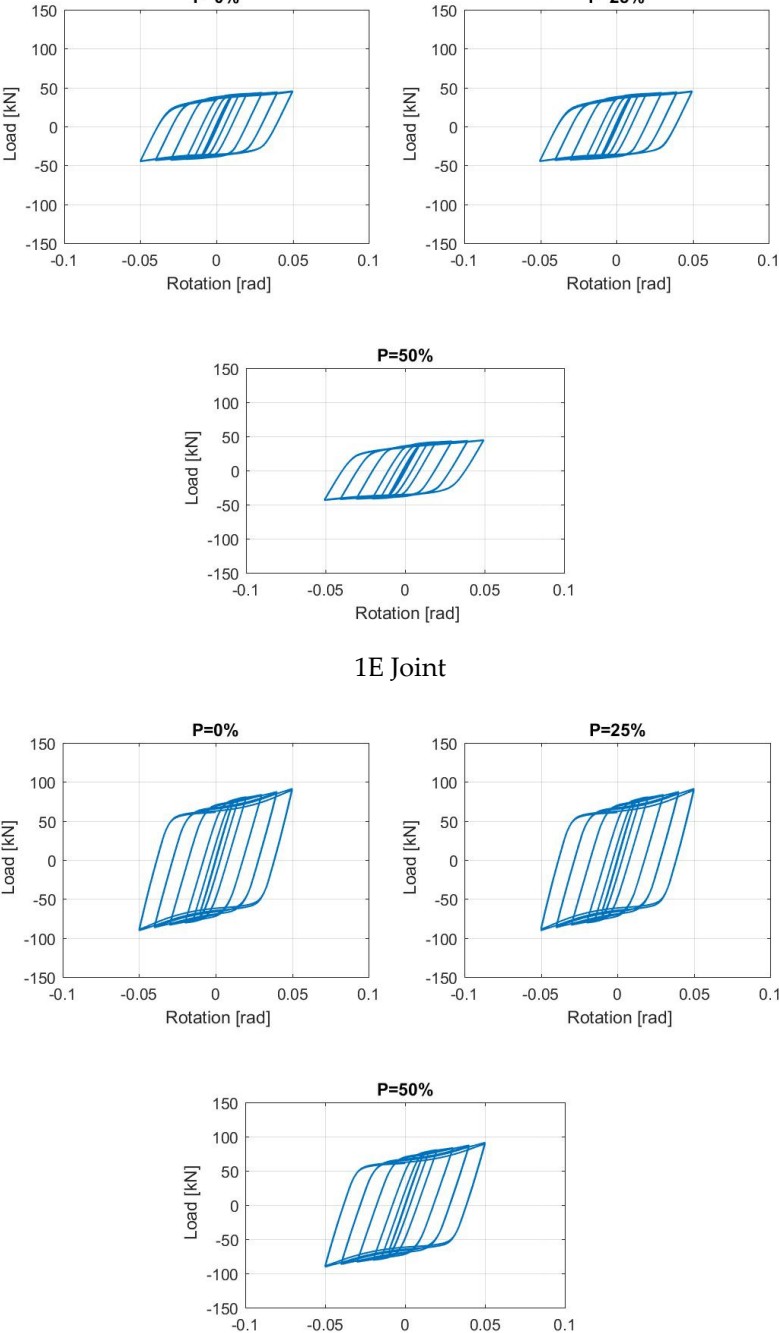

**Figure 12.** *Cont.*

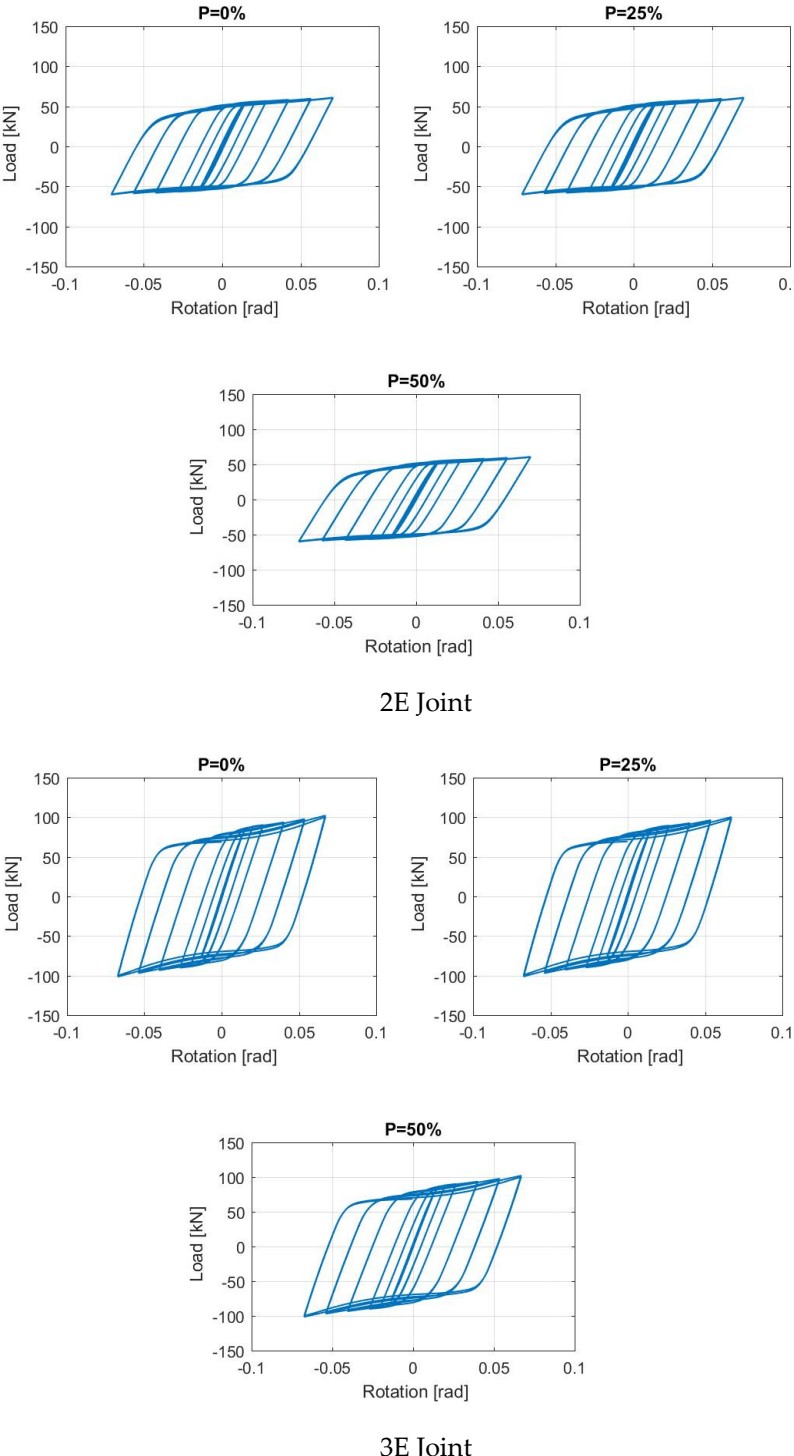

2E Joint

3E Joint

**Figure 12.** *Cont.*

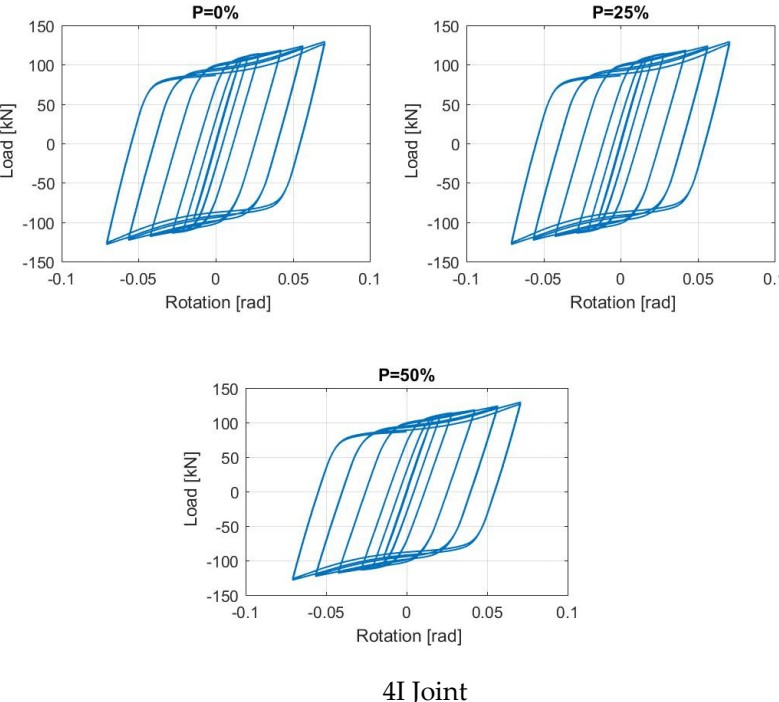

4I Joint

**Figure 12.** Summary of Load-displacement curve according to Equivalent Load Method in joints. Note: P, is the axial load applied in numerical model, Py = FyAg, where Fy is the plastic modulus and Ag is the gross area.

Figure 13, shows the normalized tangent stiffness (slope for each loop in load/reload segment for elastic loop) versus rotation in joints studied. Values close to 1 were obtained for all rotation levels. Consequently, no degradation of tangent stiffness is observed. Likewise, in Figure 14 the normalized secant stiffness (slope of the line that joins a point of maximum load with the origin/slope for elastic loop) curves are reported. A 30% to 40% of stiffness can be sustained for 4% drift and similar degradation pattern for all joint configurations was reached.

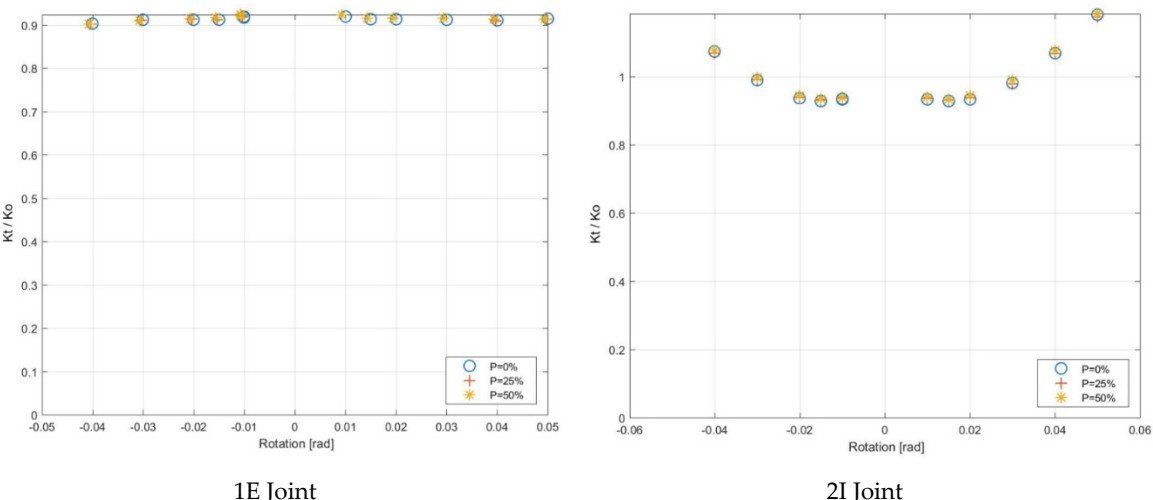

1E Joint                          2I Joint

**Figure 13.** *Cont*.

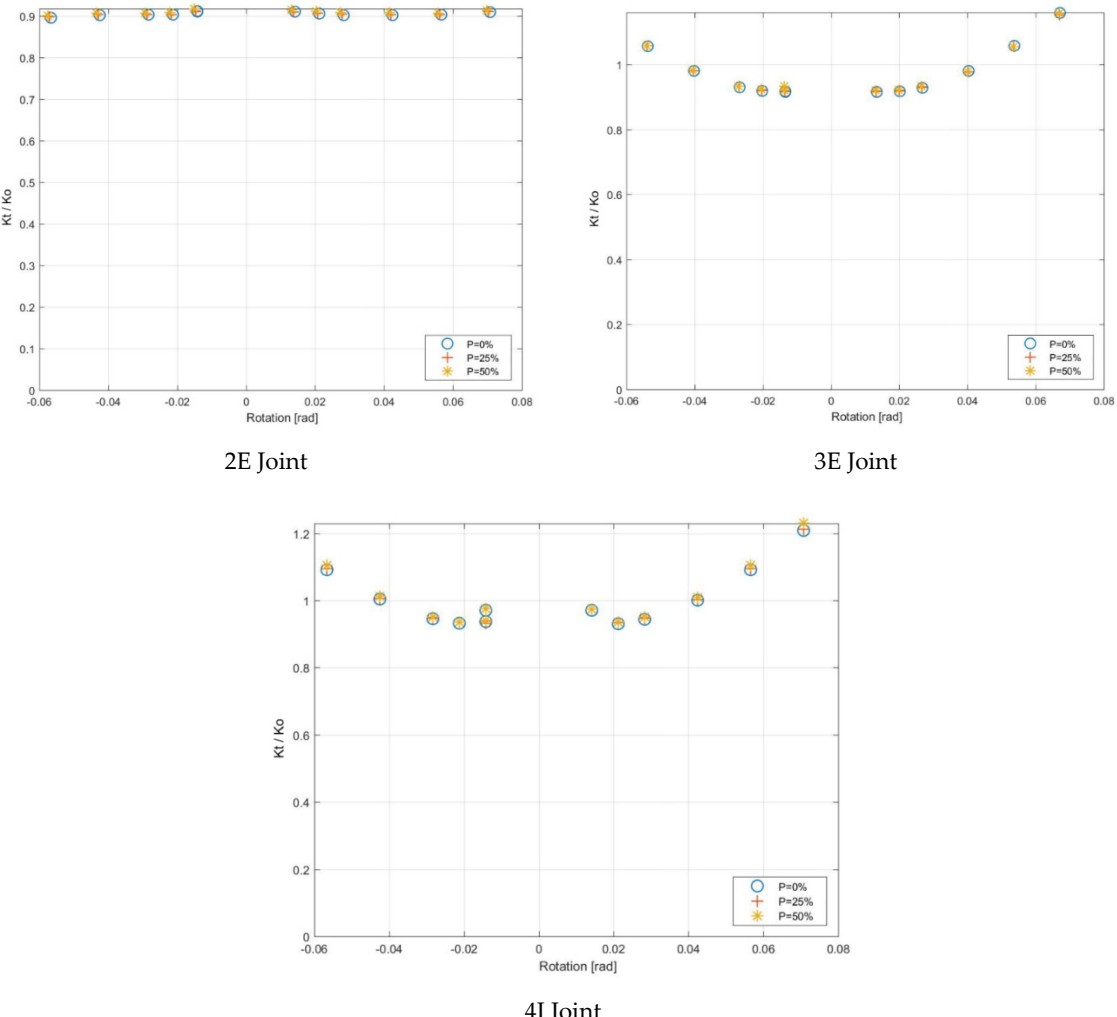

2E Joint

3E Joint

4I Joint

**Figure 13.** Summary of Tangent Stiffness vs. Rotation according to equivalent load-displacement Method.

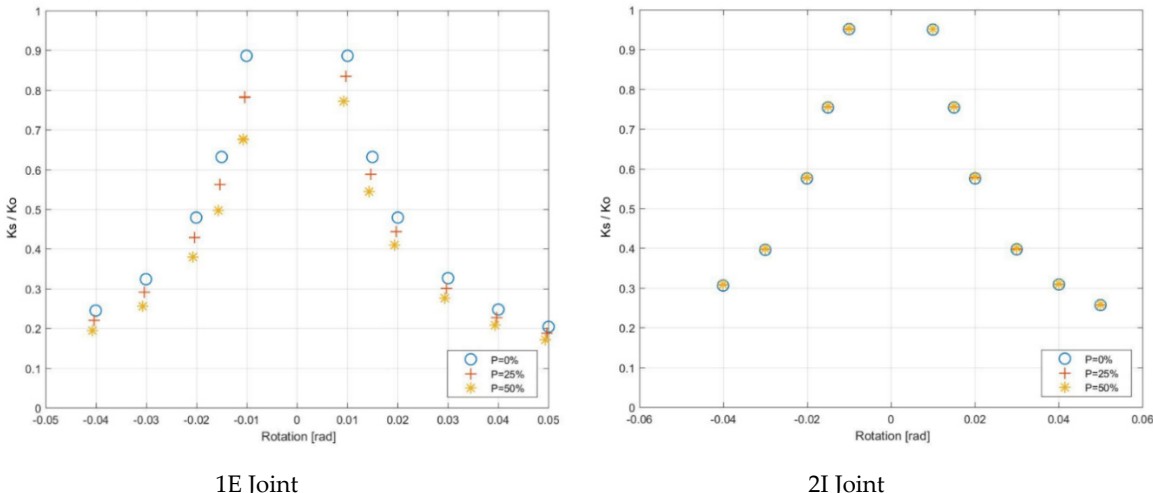

1E Joint

2I Joint

**Figure 14.** *Cont*.

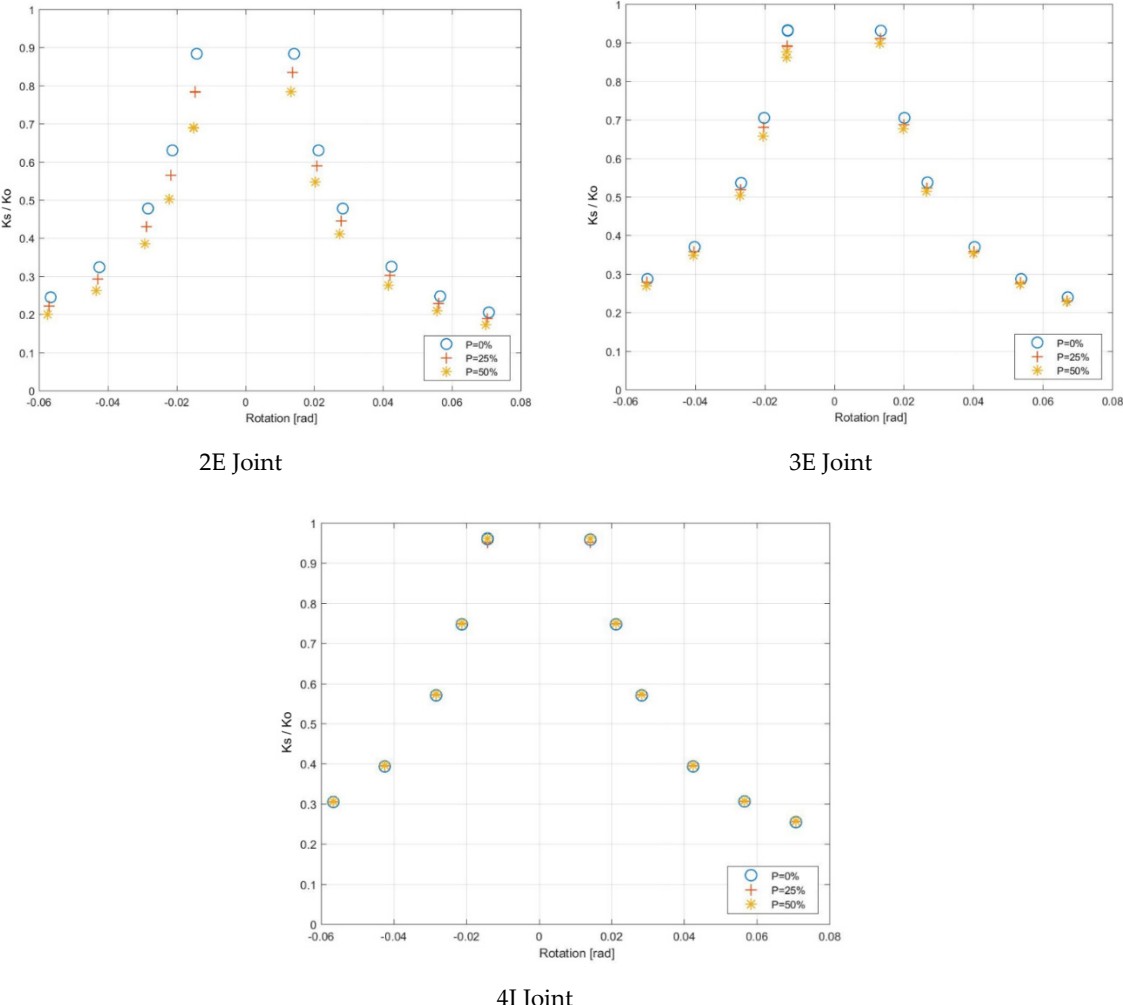

2E Joint
3E Joint
4I Joint

**Figure 14.** Summary of Secant Stiffness vs. Rotation according to equivalent load-displacement Method.

As shown in Figure 15, a higher dissipated energy was obtained in joints with two or more elements. The 2I joint dissipate 25% more than 2E joint despite having equal number of beams. Such as was observed, the dissipated energy is significant from 2% drift, due to elastic behavior of joints. Respect to equivalent damping, a 5% is obtained for values close to 2% drift in all joints (see Figure 16). These values are coherent with the common practice in design of steel buildings.

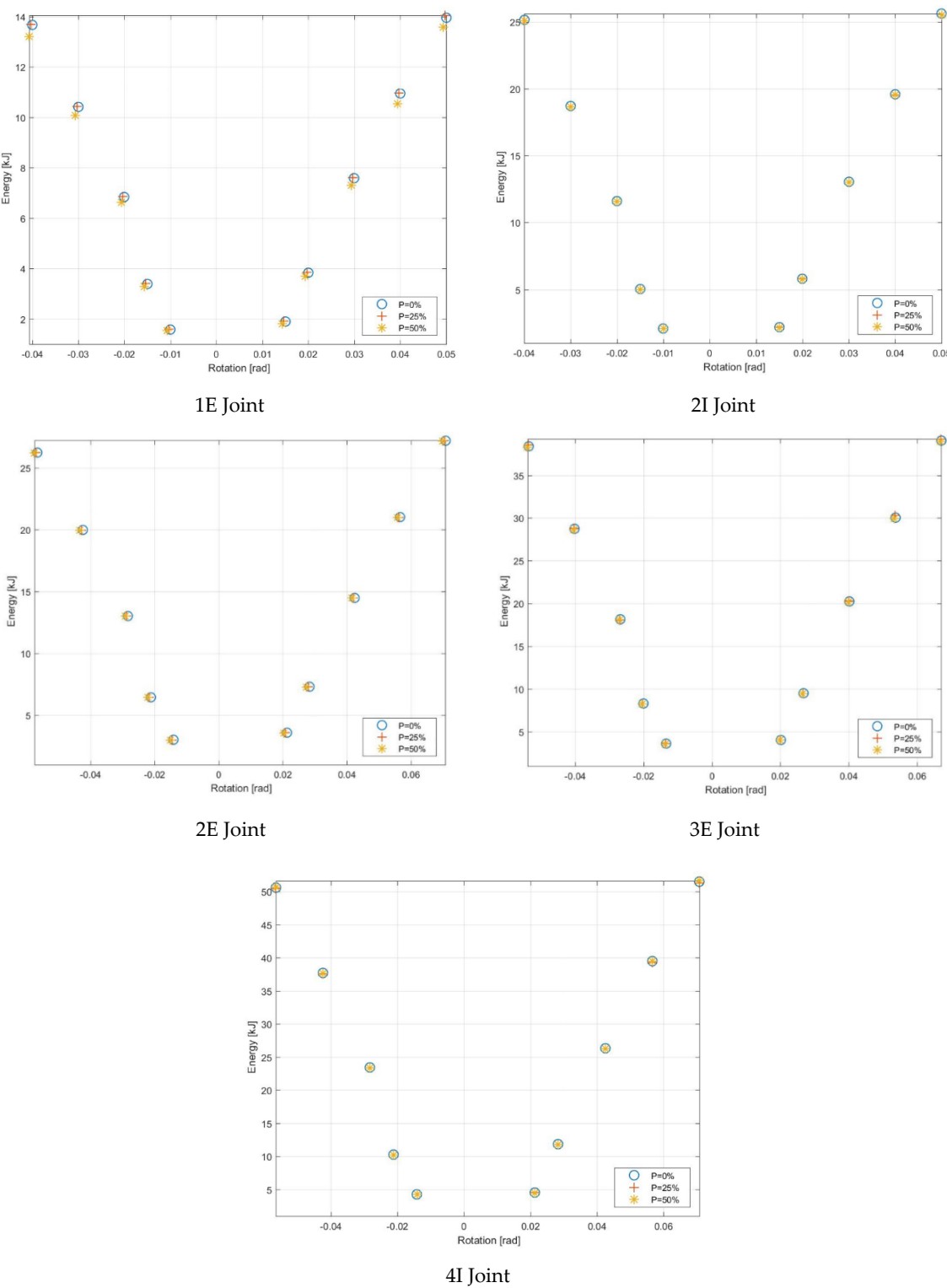

1E Joint

2I Joint

2E Joint

3E Joint

4I Joint

**Figure 15.** Summary of Dissipated Energy vs. Rotation according to equivalent load-displacement Method.

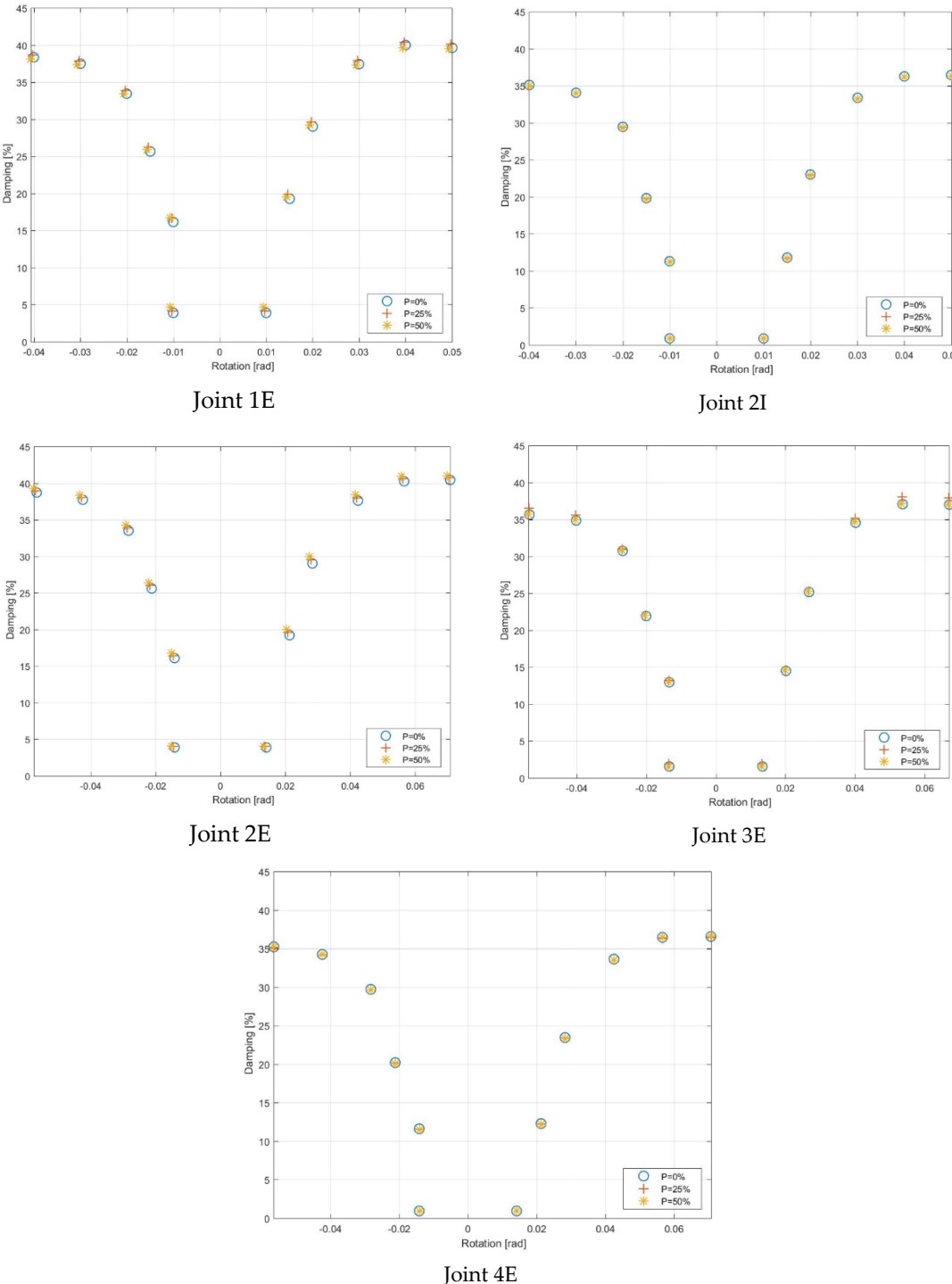

**Figure 16.** Summary of Equivalent Damping vs. Rotation according to equivalent load-displacement Method.

Although the column axial load level is one significant consideration, the results obtained show that its effect on the connection behavior is not critical, such as was observed in hysteresis curves. A comparison of 2D joints configurations studied and tests reported in [12] is reported in Table 4. A similar rotation level and dissipated energy was obtained between 1E Joint and Tests performed in [12].

**Table 4.** Summary of equivalent load-displacement results.

| Joint | P/Py [%] | Max. Load [kN] | Max. Rotation [rad] | Dissipated Energy [kJ] |
|---|---|---|---|---|
| Test 1 | 0 | 65.26 | 0.06 | 9.6 |
| Test 2 | 0 | 71.9 | 0.05 | 11 |
| Test 3 | 0 | 70.7 | 0.05 | 11.9 |
| | 0 | 45.43 | 0.05 | 13.8 |
| 1E | 25 | 45.33 | 0.05 | 13.8 |
| | 50 | 44.72 | 0.049 | 13.1 |
| | 0 | 91.24 | 0.05 | 25.8 |
| 2I | 25 | 91.21 | 0.05 | 25.8 |
| | 50 | 91.22 | 0.05 | 25.8 |
| | 0 | 60.97 | 0.071 | 26.2 |
| 2E | 25 | 60.88 | 0.07 | 26.2 |
| | 50 | 60.75 | 0.07 | 26.2 |
| | 0 | 102.18 | 0.067 | 39 |
| 3E | 25 | 100.30 | 0.067 | 39 |
| | 50 | 102.13 | 0.067 | 39 |
| | 0 | 129.28 | 0.071 | 50.8 |
| 4I | 25 | 129.24 | 0.071 | 50.8 |
| | 50 | 129.36 | 0.071 | 50.8 |

## 4. Conclusions

In this research, the cyclic performance of EP-HSS moment connection subjected to bidirectional loads was studied. A numerical study of 2D and 3D joints based on finite element models was conducted to evaluate the seismic performance of I-beams to HSS-columns. Based on the cyclic loading results of 15 joints configuration, the following conclusions can be drawn:

(1) The failure modes were mainly concentrated in the beams which complies with the requirements established in AISC seismic provisions. A combined failure mode is not obtained in either joint. A moment resistance and drift greater than 0.8 Mp and 4%, respectively, is achieved in 2D and 3D joints.

(2) The axial load is not critical for the EP-HSS moment connection. However, it affects the resistance of the columns due to the increased stresses in the wall of tubular columns.

(3) Compared with 2D joints, 3D joints can reach higher deformations even when a similar number of beams is used. Therefore, 3D effect in joints shall be considered and bidirectional effect is critical in the seismic response of joints.

(4) The pattern of dissipated energy and equivalent damping is similar in 2D and 3D joints, however, due to number of beams, the 3D joint dissipates more than 2D joints. A degradation of stiffness in joints is not obtained, because of the confinement provided by the external diaphragms to the column panel. Additionally, vertical and horizontal diaphragms allow one to resist the expected moment of the beam without damage in the column.

**Author Contributions:** Conceptualization, methodology, writing—original draft preparation, visualization and supervision, E.N.; software, R.L., E.N.; validation and investigation, E.N., R.H.; formal analysis and Furthermore, R.L.; writing—review and editing, R.H., E.N. All authors have read and agreed to the published version of the manuscript.

**Funding:** This research received no external funding.

**Conflicts of Interest:** The authors declare no conflict of interest.

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
