# Peer review of "Cyclic Performance of End-Plate Biaxial Moment Connection with HSS Columns"

_metals, doi:10.3390/met10111556_

Round 1
Reviewer 1 Report
The paper regards FE studies on an interesting connection typology with HSS. The paper is rather well organized and useful for the readers of the journal. The topic perfectly fits into the scopes of the journal. In my opinion, the paper could be accepted in its present form.
Author Response
November 09, 2020
Metals, Special Issue "Advances in Structural Steel Research"
MDPI
Subject: Revised Manuscript “metals-962286”
Dear Editorial Staff:
Please find enclosed the updated version of the paper entitled Cyclic performance of endplate biaxial moment connection with HSS columns by Eduardo Nuñez, Roberto Lichtemberg and Ricardo Herrera. This draft has been extensively revised to address the comments received following the initial review of the paper. The authors would like to thank the reviewers for their comments, which were very helpful in improving the quality and clarity of the manuscript. Attached is a summary of how the recommendations of the reviewers were addressed in the revised version of the paper.
Should you need to contact me, please contact me via email at enunez@ucsc.cl
Sincerely,
Eduardo Nuñez, PhD
Department of Civil Engineering
Universidad Católica de la Santísima Concepción, Concepción, Chile

Reviewer 2 Report
General
This paper presents numerical study on 2D and 3D connections between steel I-beams and HSS columns using end plates. The study uses a refined FEM numerical model and a large number (15) of 2D and 3D joint configurations; this is the strong point of the paper. The main weak points are: (i) the numerical model is not calibrated with experimental results, and (ii) a more in-depth discussion of the numerical results (not only a description) is needed. This includes design implications and comparison with other types of connections referred in the Introduction section. Relevant information such as the bilinear loading protocol applied. The loading protocol applied play an important role on the results. Finally, the quality of many plots is bad (see below). In the opinion of this reviewer, before possible publication the paper should be completed and extended as noted below; several concerns and clarifications should be addressed.
Particular
1.-Line 33-34. Explain the sentence “this induces the use of only a few elevations on each direction to resist lateral loads…”. Revise if the term “induces” is appropriate. The authors mean that only a low number of stories can be built with this solution?
2.-Line 67. What does it mean that the design of the end-plate is optimized 16%? What is the meaning of this percentage?
3.-Line 99. The authors should explain in detail what is new in the paper under review with respect to the numerical and experimental study published by the authors in [12]. The authors should clearly specify the objective of the study.
4.-Line 112. Provide some information of the building; e.g. number of stories, bays, spans, the seismic loads considered, to which story does these connections belong and so on.
5.-Lines 114. Detail the ratio of the sum of moment capacity of columns to beams of each joint and relate this value with the minimum value required by relevant codes (i.e. Eurocodes and US code).
6.-Line 122. Explain in some detail the “equivalent load-displacement method” and provide references the first time it is mentioned in the text.
7.-Line 139. Specify if zero moment points in columns are assumed at mid height.
8.-Line 149. Provide the reference for the “experimental conditions”.
9.-Section 2.1. Only the loading protocol applied to model 1E is described. How is the loading protocol (Table 1) applied to the beams in case of connections with more than one beam?.
10.-Include a plot with the constitutive law (stress strain) adopted for steel and explain the hysteretic behaviour under cyclic loading.
11.-Line 172. Include the equations used to convert the material strain-stress curve obtained from coupon test to true stress and true strain.
12.-Section 3. The authors say in the abstract that this study is a continuation of a previous experimental investigation. They should demonstrate first that the numerical model used is well calibrated with experimental results. To this end, they should include some plot comparing the force-displacement curve obtained from at least one test (specimen 1E?) with the prediction provided by the numerical model; besides the force-displacement curves, the authors should compare also the amount of dissipated energy by the specimen and by the numerical model.
13.-Line 201. Define the tangent stiffness. The authors mean the slope of the unloading branch?. Further, Figure 11 inlcudes a Figure with the energy plot in the vertical axis instead of stiffness (joint 2I).
14.-Line 212. Define the secant stiffness
15.-Line 222. Describe (including the equation), how the equivalent damping is calculated.
Miscellaneous
1.-Line 105. Revise English. As show? Or As shown?
2.-The quality of Figures 3 and 4 is very bad. Replace the figure and increase the size of the text.
3.-Line 124. The c of Vc should be a subindex.
4.-Line 142. What does it mean “…bolts in slip critical”?
5.-Explain the meaning of the arrows in Fig. 5a.
6.-The quality of figures 7 and 10 is bad. Further, the size of the plots and of the text should be increased.
7.-The quality of Figures 11 and 12 is very bad. The text is extremely small.
8.-Line 210. Use the subindex (i.e. the t in Kt).
Author Response

(The authors gave the same response as above.)

Reviewer 3 Report
Good work.
Please explain the abbreviation HSS on Line 39.
From Line 95, in Introduction, it is shown what and how has been researched. Unfortunately, the Introduction looks really heavy. Please take part of the description of how it was examined from Introdiction in Modeling (Section 2).
Line 139. story = case
Figures 7, 10 and other. Please explain the abbreviation P
Figures 8, 9 and other. The texts are difficult to read.
Author Response

(The authors gave the same response as above.)

Reviewer 4 Report
The manuscript dealt with a practical problem regarding the cyclic performance of end-plate moment connection between I-beam to HSS column using the finite element method. Overall, the topic could interest some readers, and the numerical analysis could serve as a good tool to evaluate the performance of the moment connection designs. However, the authors could have improved the quality of the manuscript and engage readers by providing more details of their numerical analysis. In addition, the authors should consider validating the numerical method by comparing the numerical results with some experimental data. It is hard to make convincing statements by relying on purely numerical evidence without even validating the numerical methods since many of the assumptions have been made in the numerical models. By the way, the quality of the figures in the manuscript is very bad, making it very hard to recognize the symbols and numbers. It is not related to the quality of writing, but the authors should seriously consider providing high-quality figures. Some comments as follows for the authors’ consideration.
(1) It is advised to provide a more detailed overview of the current design code, construction practice, and state of the art research regarding steel moment resisting framed structures in Section 1.
(2) Figure 5(a) was supposed to illustrate the boundary conditions adopted in the finite element modes. However, it is not intuitive, and the readers cannot understand how the boundary conditions were applied without referring to the text.
(3) It is confusing that both frictional and frictionless contacts were mentioned from line 157 to line 160 but it was not specified the contacts surfaces on which the frictional contact algorithm was applied.
(4) It was mentioned that the solid elements and beam elements were used. The authors referred to Figure 6 for more information, but one cannot tell which portions were modeled with solid or beam elements. Please specify. In addition, how did the authors deal with the connection between solid and beam elements?
(5) Please specify the hardening law, the associated parameters, and elastic constants of the adopted materials models in Section 2.2 for reproducibility of the results.
(6) Figure 7 showed the results of normalized moment vs rotation curves. Please specify how was the moment calculated and what quantity was used to normalize it.
(7) It is confusing that the authors mentioned that no degradation of stiffness is observed in line 188. As described in the model setup, plasticity is the only inelastic deformation mechanism incorporated and not damage or mechanisms which would cause stiffness degradation was mentioned. Why did the authors expect that there may be stiffness degradation observed in the simulation results? It is understandable that there could be stiffness degradation in actual experimental data, but it cannot appear automatically if the authors did not consider it in the model.
(8) The meaning of resistance mentioned in line 188 is also confusing. Please explain.
(9) The authors need to explain why they saw what they saw in Figure 10 instead of just describing the observations in the results (line 202 to 207). A short description of the so-called equivalent load method is also needed to engage the readers.
(10) Unless the reviewer misunderstood the meaning of tangent stiffness, the representation of Figure 11 is misleading as explained in (7). By the way, the subfigure for Joint 2I was misplaced.
(11) Line 219-223: the authors should describe how the equivalent damping quantities were calculated. It is not as intuitive as dissipated energy.
Author Response

(The authors gave the same response as above.)

Round 2
Reviewer 2 Report
The authors have addresses the concerns raised by this reviewer.
Author Response
Metals, Special Issue "Advances in Structural Steel Research"
MDPI
Subject: Revised Manuscript “metals-962286”
Dear Editorial Staff:
The authors would like to thank the reviewers for their comments, which were very helpful in improving the quality and clarity of the manuscript.
Sincerely,
Eduardo Nuñez, PhD
Department of Civil Engineering
Universidad Católica de la Santísima Concepción, Concepción, Chile
Reviewer 4 Report
The quality of the manuscript has been improved. One last comment: please include a legend in Fig. 7.
Author Response
November 11, 2020
Metals, Special Issue "Advances in Structural Steel Research"
MDPI
Subject: Revised Manuscript “metals-962286”
Dear Editorial Staff:
Please find enclosed the updated version of the paper entitled Cyclic performance of end-plate biaxial moment connection with HSS columns by Eduardo Nuñez, Roberto Lichtemberg and Ricardo Herrera. This draft has been extensively revised to address the comments received following the initial review of the paper. The authors would like to thank the reviewers for their comments, which were very helpful in improving the quality and clarity of the manuscript. Attached is a summary of how the recommendations of the reviewers were addressed in the revised version of the paper.
Should you need to contact me, please contact me via email at enunez@ucsc.cl
Sincerely,
Eduardo Nuñez, PhD
Department of Civil Engineering
Universidad Católica de la Santísima Concepción, Concepción, Chile
